



# Estimation of $NO_x$ and $SO_2$ Emissions from Sarnia, Ontario using Mobile-MAX-DOAS and a $NO_x$-Analyzer

Zoe Y. W. Davis[1], Sabour Baray[2], Chris A. McLinden[3], Aida Khanbabakhani[2], William Fujs[2], Csilla Csukat[2], Jerzy Debosz[4], Robert McLaren[2].

[1]Graduate Program in Earth and Space Science, York University, Toronto, M3J 1P3, Canada
[2]Centre for Atmospheric Chemistry, York University, Toronto, M3J 1P3, Canada
[3]Environnment and Climate Change Canada, Toronto, M3H 5T4, Canada
[4]Air Quality Monitoring and Assessment Unit, Ontario Ministry of the Environment, Conservation and Parks, Etobicoke, M9P 3V6, Canada

*Correspondence to*: Zoe Y. W. Davis (zoeywd@yorku.ca) or R. McLaren (rmclaren@yorku.ca)

**Abstract.** Sarnia, ON experiences pollutant emissions disproportionate to its relatively small size. The small size of the city limits traditional top-down emission estimate techniques (e.g., satellite) but a low-cost solution for emission monitoring is Mobile-MAX-DOAS. Measurements were made using this technique from 21/03/2017 to 23/03/2017 along various driving routes to retrieve vertical column densities (VCDs) of $NO_2$ and $SO_2$ and to estimate emissions of $NO_x$ and $SO_2$ from the Sarnia region. A novel aspect of the current study was the installation of a $NO_x$ analyzer in the vehicle to allow real time measurement and characterization of near-surface $NO_x/NO_2$ ratios across the urban plumes, allowing improved accuracy of $NO_x$ emission estimates. Confidence in the use of near-surface measured $NO_x/NO_2$ ratios for estimation of $NO_x$ emissions was increased by relatively well-mixed boundary layer conditions. These conditions were indicated by similar temporal trends in $NO_2$ VCDs and mixing ratios when measurements were sufficiently distant from the sources. Leighton ratios within transported plumes indicated peroxy radicals were likely disturbing the $NO$-$NO_2$-$O_3$ photostationary state through VOC oxidation. The average lower limit emission estimate of $NO_x$ from Sarnia was $1.60 \pm 0.34$ tonnes hr$^{-1}$ using local 10 m elevation wind-speed measurements. Our estimates were larger than the downscaled annual 2017 NPRI reported industrial emissions of 0.9 tonnes $NO_x$ hr$^{-1}$. Our lower limit estimate of $SO_2$ emissions from Sarnia was $1.81 \pm 0.83$ tonnes $SO_2$ hr$^{-1}$, equal within uncertainty to the 2017 NPRI downscaled value of 1.85 tonnes $SO_2$ hr$^{-1}$. Satellite-derived $NO_2$ VCDs over Sarnia from the Ozone Monitoring Instrument (OMI) were lower than Mobile-MAX-DOAS VCDs, likely due to the large pixel size relative to the city's size. The results of this study support the utility of the Mobile-MAX-DOAS method for



estimating $NO_x$ and $SO_2$ emissions in relatively small, highly industrialized regions especially when supplemented
with mobile $NO_x$ measurements.

**1 Introduction**

Differential Optical Absorption Spectroscopy (DOAS) is a remote sensing technique that quantifies tropospheric
trace-gases using light spectra and the unique spectral absorption cross sections of trace-gases. DOAS has been used
since its introduction by (Platt et al., 1979) to measure small molecular species including $NO_2$, $SO_2$, OH, BrO, $NO_3$,
$NH_3$, ClO and others. One advantage of the technique is the potential for simultaneous quantification of multiple
trace-gases (e.g., $SO_2$ and $NO_2$) (Platt et al., 2008). The Multi-Axis DOAS (MAX-DOAS) method allows sensitive
quantification of tropospheric pollutants by measuring scattered sunlight spectra at multiple viewing directions
and/or elevation angles. Spectra measured at elevation angles close to horizon-pointing have high sensitivity to
ground-level gases since the light paths are longer near the surface (Honninger et al., 2004). Ground-based MAX-
DOAS measurements quantify total boundary layer pollution loading by determining tropospheric vertical column
densities (VCDs) of trace-gases. These measurements are, therefore, well suited to measurement of total emissions
into an air mass. VCDs are independent of boundary layer height, unlike mixing ratios, and are spatially averaged
(horizontally and vertically) on the order of a few kilometres along the light path. Ground-based MAX-DOAS can
also retrieve vertical profiles of aerosol extinction and trace-gases by combining MAX-DOAS data with radiative
transfer modelling  (Friess et al., 2006; Heckel et al., 2005; Honninger et al., 2004; Honninger and Platt, 2002; Irie et
al., 2008; Wagner et al., 2004, 2011).
The recently developed Mobile-MAX-DOAS technique allows measurement of trace-gas emissions from a region of
interest by driving the instrument around the region. The method can estimate emissions on a nearly hourly basis in
a region with a spatial resolution of ~1 km. Mobile MAX-DOAS has been used to estimate $NO_x$ emissions from a
shipping and industrial areas (Rivera et al., 2010), power-plants (Wu et al., 2017) and cities (Ibrahim et al., 2010;
Shaiganfar et al., 2011, 2017), validate satellite and air quality modelled VCDs (Dragomir et al., 2015; Shaiganfar et
al., 2015), estimate surface $NO_2$ mixing ratios from $NO_2$ VCDs (Shaiganfar et al., 2011), and determine the
horizontal variability of trace-gas VCDs within satellite pixels (Wagner et al., 2010). Mobile-MAX-DOAS is a "top-
down" approach for quantifying real-world emissions that can be used to validate "bottom-up" emission inventories
(Shaiganfar et al., 2011).





Sarnia, Ontario, a small Canadian city, experiences pollutant emissions due to a large number of industrial chemical
and oil processing facilities, vehicular exhaust from the Canada-U.S.A. international border crossing, emissions
from large ships travelling through the St Clair River, vehicular traffic, residential heating and other anthropogenic
emissions from the city populace, and transnational air pollution from Ohio, Illinois and Michigan (Oiamo et al.,
2011). These sources contribute to increased levels of air pollutants such as $NO_x$, VOC's and $SO_2$, which are
precursors of $PM_{2.5}$ and $O_3$ (Ministry of the Environment and Climate Change, 2015). Traditional "top-down"
methods for quantifying pollutant emissions from small cities (e.g., satellite monitoring, aircraft studies) are limited
by the small footprint. Additionally, in-situ air quality monitoring stations are limited by the bias towards near-
surface emissions and under-sampling of elevated emissions (Tokarek et al., 2018).
The Mobile-MAX-DOAS method has advantages over satellite, aircraft and in-situ techniques. Major advantages
over satellite techniques include 1) emissions can be estimated without the need for an a-priori vertical profile, 2)
accuracy of estimates can increase rather than decrease for smaller source regions, and 3) emissions may be
estimated many times per day. Satellite retrievals are useful for estimating "top-down" emissions on regional and
global scales over long periods of time (Huang et al., 2014; Kim et al., 2014; Liu et al., 2016; McLinden et al.,
2012). However, accuracy over small regions can be limited by insufficient pixel resolution due to horizontal
averaging and retrieval reliance on modelled a-priori vertical profiles that may not resolve small regions (Heckel et
al., 2011). Aircraft studies can quantify emissions from cities but are relatively expensive. Major advantages of the
Mobile-MAX-DOAS method over aircraft techniques (Baray et al., 2018; Gordon et al., 2015) are that 1) MAX-
DOAS VCDs are already vertically integrated, reducing the uncertainties due to interpolation of measurements at
multiple flight altitudes and 2) MAX-DOAS studies are logistically easier to conduct. The Mobile-MAX-DOAS
technique is a solution for quantifying pollutant emissions that complements the aforementioned techniques as well
as in-situ monitoring, through the ability to observe localized surface based and elevated emissions.
An uncertainty associated with MAX-DOAS and satellite methods when estimating $NO_x$ emissions from $NO_2$
measurements is the assumptions concerning the $NO_x/NO_2$ relationship in the air mass, which can be variable both
spatially and temporally. The $NO_x/NO_2$ ratio is often assumed to be spatially constant, taken from literature based on
the season,estimated using atmospheric modelling or occasionally taken from aircraft measurements when available
(Rivera et al., 2010). In this study, we combined the Mobile-MAX-DOAS method with simultaneous mobile $NO_x$



measurements (NO, $NO_2$, $NO_x$) to increase knowledge of the $NO_x/NO_2$ ratio in the air mass spatially and temporally
in order to improve the accuracy of the $NO_x$ emission estimates obtained from $NO_2$ measurements. A stationary
modular meteorological station was deployed in the airshed provided auxillary meteorological information, typically
a major source of uncertainty in Mobile-MAX-DOAS emission estimations. Hourly wind data measured at 10 m
elevation (agl) were also available from local, permanent monitoring stations. Vertical wind profiles were modelled
in high resolution (1 km x 1 km) using the version 3.9.1 Weather Research and Forecasting model (WRF) centred on
Sarnia ($42.9745^o$ N, $82.4066^o$ W) in an attempt to improve upon emissions values calculated using near-surface
wind-speed, since wind-speeds are expected to increase with altitude. However, inter-comparison of WRF modelled
winds with measured near-surface winds during the study period indicated poor model performance (see
Supplement S2.2 for detailed results). Emissions in this study were therefore calculated using the 10 m measured
winds to provide lower limit estimates of the hourly emissions.
Our study objectives were to 1) examine the relationship between the $NO_2$ near-ground mixing ratios and the $NO_2$
tropospheric VCDs, 2) determine $NO_x$ and $SO_2$ emissions from the city of Sarnia including industrial sources, 3)
determine the impact of $NO_x/NO_2$ variability on the accuracy of $NO_x$ emission estimates, and 4) examine OMI
satellite intrapixel $NO_2$ homogeneity. This study aims to demonstrate the utility of this method for determining
trace-gas emissions and monitoring pollutant transportation in Sarnia and similar urban/industrial areas.
**2 Experimental**
**2.1 Location and Instruments**
Measurements were conducted in and around the city of Sarnia (42.9745° N, 82.4066° W), located in southwestern
Ontario, Canada at the border with Port Huron, MI, U.S.A (Fig.1). The routes driven in the vehicle aimed to capture
major $NO_x$ and $SO_2$ emission sources at different distances downwind, dependent on the prevailing wind conditions.
The metro area has a population of ~72,000 (2016 census) and an area of ~165 $km^2$. Sources of air pollution in this
region include emissions from large ships, anthropogenic emissions from the cities of Sarnia and Port Huron,
transport from the cities of Windsor and Detroit (60 km SW), the St Clair and Belle River power-plants (20 km
SSW), oil refineries and chemical industry in Sarnia, and the cross-border traffic between Canada and the U.S.A.



along Highway 402. Emissions from ships along the St. Clair River, normally a major source, were absent during
the time of our study since the canal had not opened for the season.
A mini-MAX-DOAS instrument (Hoffmann Messtechnik GmbH) measured scattered sunlight spectra during three
days: 21/03/2017 to 23/03/2017 ("Days 1 to 3") while mounted on top of a car in a backwards pointing direction.
The instrument has a sealed metal box containing entrance optics, UV fibre coupled spectrometer and electronics.
Incident light is focused on a cylindrical quartz lens (focal length = 40 mm) into a quartz fibre optic that transmits
light into the spectrometer (OceanOptics USB2000) with a field of view approximately $0.6^{o}$. The spectrometer has a
spectral range of 290-433 nm, a 50µm wide entrance slit yielding a spectral resolution was ~0.6 nm. The
spectrometer is cooled and stabilized by a Peltier cooler. Spectrometer data was transferred to a laptop computer via
USB cable. Spectra were obtained with an integration time of ~1 minute with the continuously repeating sequence
of viewing elevation angles ($30^{o}$, $30^{o}$, $30^{o}$, $30^{o}$, $40^{o}$, $90^{o}$). The vehicle was driven at a low but safe target speed of 50
km hr$^{-1}$ when possible to provide a spatial resolution of ~ 1 km, but speeds were occasionally up to 80 km hr$^{-1}$ when
necessary. Tropospheric VCDs were estimated from the $30^{o}$ and $40^{o}$ elevation angle spectra. The $40^{o}$ spectra allow
verification that aerosol levels were sufficiently low to determine VCDs without radiative transfer modelling since
VCDs obtained from both angles should be equal within ±15% under low to moderate aerosol loading conditions
(Wagner et al., 2010). The cool temperatures in March aided in this as secondary organic aerosol loading tends to be
low in this season due to an absence of biogenic emissions.
A Model 42 chemiluminescence NO-NO$_2$-NO$_x$ Analyzer (Thermo Environmental Instruments Inc.) mounted in the
vehicle measured NO, NO$_2$, and NO$_x$ (NO+NO$_2$) near-surface mixing ratios. A PTFE inlet tube (5m length and
ID=1/4") was mounted above the front vehicle window on the passenger side (~1.5 m above ground). The
instrument alternately recorded average NO-NO$_2$-NO$_x$ mixing ratios with a temporal resolution of 1 minute. Most of
the routes were driven downwind of Sarnia on rural remote roads with little to no traffic such that NO$_x$ emissions
from other vehicles were not a concern. When NO$_x$ from other vehicles was a potential concern, data was filtered
out via careful note taking. The instrument indirectly measures NO$_2$ by subtracting the NO chemiluminescence
signal obtained when air bypasses a heated Molybdenum (Mo) convertor from the successive total NO$_x$
chemiluminescence signal obtained when air passes over the Mo-convertor. The NO$_x$ analyzer can overestimate NO$_x$
and NO$_2$ due to the potential contribution of other non-NO$_x$ reactive nitrogen oxides (NO$_z$) other than NO$_2$ that can



also be reduced to NO by the Mo converter ($HNO_3$, HONO, organic nitrates, etc.), leading to an overestimation
(Dunlea et al., 2007). Since this overestimation is more important in low $NO_x$ regions, only data with $NO_x$ mixing
ratios > 3 ppb were used. Mixing ratios of <3ppb $NO_2$ were only measured outside of plume-impacted regions when
$NO_2$ VCDs were also low. The potential error in $NO_x/NO_2$ ratios is addressed further in section 3.2. $NO_x$ mixing
ratios can also have an error when successive NO and $NO_x$ measurements occurred in areas with a significant
temporal gradient in the $NO_x$ emissions. Such gradients were seen due to passing vehicles or localized industrial
$NO_x$ plumes. These data were removed based on records of passing vehicles and other local near-surface sources or
whenever the $NO_2$ mixing ratios were reported as negative. Few data points were removed because the routes driven
were primarily rural roads with extremely low traffic density.
Aura satellite Ozone Monitoring Instrument (OMI) data were obtained for overpasses of the Sarnia, Ontario area for
Days 1 and 3. Tropospheric $NO_2$ VCDs are the NASA Standard Product Version 3.0 with AMFs recalculated using
the Environment and Climate Change Canada regional air quality forecast model GEM-MACH. The OMI
instrument makes UV-vis solar backscatter radiation measurements with a spatial resolution of 13x24 $km^2$ at nadir
and up to 28×150 $km^2$ at swath edges (Ialongo et al., 2014). The $NO_2$ detection limit of OMI is $5×10^{14}$ molec $cm^{-2}$
(Ialongo et al., 2016). The OMI data used were screened for row anomalies that have affected OMI data since June
2007 (Boersma et al., 2007).

**152 2.2 MAX-DOAS Determination of VCDs**

Trace-gas Differential Slant Column Densities (DSCDs) were obtained using the DOAS technique (Platt et al.,
2008) with the spectral fitting range of 410-435 nm for $NO_2$ at 293 K and 307.5-318 nm for $SO_2$ at 293 K. All trace-
gas cross-sections used were from (Bogumil et al., 2003). For both gases, spectral fits also included a Fraunhofer
Reference Spectrum (FRS), Ring Spectrum created from the FRS, $O_3$ cross-sections at 223 K and 297 K, and a
third-order polynomial. The $NO_2$ cross-section was included in the $SO_2$ fits. $NO_2$ DSCDs from Day 1 were fit
against a single, same-day FRS obtained in a low-pollutant region near solar-noon time. These DSCDs were
corrected for SCD(FRS) and SCD(Solar Zenith Angle (SZA)) contributions using the $DSCD_{offset}$ method (Wagner et
al., 2010). The SCD(FRS) is the constant tropospheric trace-gas SCD component present in the FRS that causes an
underestimation in the fitted DSCD. The SCD(SZA) is the difference between the stratospheric trace-gas component
in the FRS and the measured non-zenith spectra. SCD(SZA) varies over time of day ($t_i$), maximizing overestimation





in the DSCD early and late in the day. The sum of SCD(FRS) and SCD(SZA) is collectively known as the
$DSCD_{offset}$. The $DSCD_{offset}(t_i)$ function was estimated by fitting a second order polynomial to multiple pairs of
DSCDs of spectra (non-zenith and zenith from the same sequence), described in detail in (Wagner et al., 2010).
The $DSCD_{offset}$ polynomial is most accurate when successive spectra in each sequence observe similar mixing ratio
fields, and measurements obtained many data-points over most of the daylight hours. However, routes on Days 2
and 3 included driving in and out of both high and low $NO_x$ regions within short time-periods and thus met neither
of the requirements listed above for the $DSCD_{offset}$ method. On these days, a second method was used where $NO_2$
DSCDs were fit against an FRS spectrum obtained close in time (<25 minutes) along each respective route in a low-
pollutant region. The impacts of SCD(FRS) and SCD(SZA) on the retrieved DSCDs can be assumed to be negligible
since each FRS was from a low-pollutant area and obtained close in time, respectively. This method was also used
for the Day 1 $SO_2$ route since limited data were available but included background $SO_2$ measurements close in time.
For all routes trace-gas tropospheric VCDs were determined by assuming a single scattering event occurred for each
photon such that the air-mass factor (AMF) depended only on the viewing elevation angle, $\alpha$, $AMF_{trop}(\alpha) \approx \frac{1}{\sin(\alpha)}$
(Brinksma et al., 2008)(Wagner et al., 2010). This "geometric approximation" is most valid under low to moderate
aerosol loading and has been shown to deviate from the typically more accurate radiative transfer modelling by up to
±20% under moderate aerosol loading (Shaiganfar et al., 2011). Day 1 VCDs were calculated following Eq. (1):

$$VCD_{trop} = \frac{DSCD_{meas}(\alpha, t_i) + DSCD_{offset}(t_i)}{\frac{1}{\sin(\alpha, t_i)}} \tag{1}$$

Days 2 and 3 $NO_x$ and Day 1 $SO_2$ VCDs were calculated following Eq. (2):

$$VCD_{trop} \approx \frac{DSCD_{meas}(\alpha, t_i)}{\frac{1}{\sin(\alpha, t_i)}} \tag{2}$$


**2.3 Estimating Trace-gas Emissions from MAX-DOAS VCDs**
Trace-gas emission estimates were calculated following a flux integral approximation Eq. (3):

$$E = \left[ (\sum_i \left( VCD_{outflow,i} - VCD_{influx,i} \right) w_i \, \sin(\beta_i) \, ds \right] \frac{MW}{Av} \tag{3}$$



where $VCD_{outflow,i}$ is the VCD measured at position i along the route s for distance ds, $VCD_{influx,i}$ is either the
measured influx values or the estimated background VCD value, $w_i$ is the wind-speed, $\beta_i$ is the angle between the
driving direction and the wind-direction, MW is the molecular weight of the target gas, and Av is Avogadro's
number. Transect routes were designed to observe both within and beyond emission impacted areas since routes
encircling the emission sources were often not possible. Flux integrals were calculated using portions of the
transects impacted only by the Sarnia urban/industrial plume in cases where plumes from other sources impacted the
transect (i.e., Day 1; U.S.A. power-plant plumes). In these cases, the end-points of integration were chosen
judiciously where $NO_2$ VCDs and surface mixing ratios decreased to a minimum at the edge of the Sarnia emissions.
This method assumes that the wind-field and trace-gas emission rates are constant during the time required to drive a
route. The validity of this assumption improves with decreased time for driving route completion. The Sarnia region
is ideal for this method since a small geographical area contains the majority of the emissions and is surrounded on
three sides by rural regions with low anthropogenic emissions.
A potential source of uncertainty in Mobile-MAX-DOAS emission estimates is variation in the wind fields and/or
source emission rates while driving (Ibrahim et al., 2010; Wu et al., 2017). Previous studies have estimated wind-
fields from local meteorology stations (Ibrahim et al., 2010), meteorological models (Shabbir et al., 2016;
Shaiganfar et al., 2011, 2017) or LIDAR measurements (Wu et al., 2017). In our study, wind field information was
obtained from a Modular Weather Station (Nova Lynx 110-WS-25DL-N) we deployed near one of the driving
routes at $(42.8148^{o}, -82.2381^{o})$ (Fig. 1) and from meteorological ground stations in the area (Fig. 1, Table S1, Fig.
S1). The modular weather station measured wind-speed and direction, temperature, relative humidity, and
barometric pressure at 2 m above the surface every 30 seconds. Wind data was available from the Sarnia-Lambton
Environmental Association (SLEA) LaSalle Road $(42.911330^{\circ}, -82.379900^{\circ})$ and Moore Line $(42.83954^{\circ}, -
82.4208^{\circ})$ meteorological stations that are located near the driving routes (Fig. 1). These stations were surrounded
by fallow, flat farm-land for at least 4 km on each side and thus should reflect total boundary layer for plumes
transported away from the city more than the urban stations (Fig. S1). The hourly wind-direction data from the
modular and permanent stations exhibited similar values $(\pm10^{o})$ and trends on Day 1 (Fig. S2). Wind-directions for
Days 2 and 3 were obtained by determining the angle of a vector drawn between the geographical locations of the
maximum $NO_2$ VCD enhancements and the industrial facilities expected to have emitted the plumes. These map-
determined wind-directions were consistent $(+/-10^{o})$ with the data from the station(s) closest to the driving route.



Comparison of wind-speed data on Days 2 and 3 was not possible due to a technical issue with the modular weather
station on these days.
The $NO_2$ VCD influx (background VCD) was estimated on Day 1 since measurement was impossible along the
western border of Sarnia due to the road configuration and proximity of industrial emissions. A $NO_2$ $VCD_{influx}$ =
$2\times10^{15}$ molec $cm^{-2}$ was estimated based on OMI satellite VCDs of ~1.5-3.5$\times10^{15}$ molec $cm^{-2}$ from the area east of
Sarnia that are expected to be similar to the $NO_2$ regime west of Sarnia. These pixels are expected to be unaffected
by other sources. The influx would be expected to be impacted by vehicular and residential emissions from the small
city of Port Huron, U.S.A., on the west side of the St Clair River (Fig. 1), which has limited industry but a moderate
level of commercial vehicle activity due to border-crossings. A first order emission estimate of vehicular $NO_x$
emissions from Port Huron from daily reported traffic counts results in an upper limit of $NO_2$ influx VCD of
~1$\times10^{15}$ molec $cm^{-2}$ (see Supplement S4). True influx would vary along the length of the measurement transect,
depending on what sources are upwind of the location. Halla et al. (2011) measured $NO_2$ tropospheric VCDs using
MAX-DOAS in a similar region approximately 70 km south-east of Sarnia. The observed $NO_2$ VCDs in that study
ranged from 0.01 to 1.25$\times10^{16}$ molec $cm^{-2}$ with a median value of $2\times10^{15}$ molec $cm^{-2}$, which is expected to be
representative of background $NO_2$ columns in this region. The highest VCD in that study was attributed to the
transport of industrial emissions from the Sarnia area and/or from Detroit, MI to the northwest and west of the site
respectively (Halla et al., 2011). Based on the range of VCDs from literature, vehicular emission estimates and
satellite measurements, a background VCD of $2\times10^{15}$ molec $cm^{-2}$ is a reasonable estimate, and emissions sensitivity
tests were conducted using influx VCDs of 0.5-3$\times10^{15}$ molec $cm^{-2}$ (Supplement S5). In contrast, the $NO_2$ $VCD_{influx}$
on Days 2 and 3 and $SO_2$ $VCD_{influx}$ on Day 1 were determined from the average VCDs measured in the low-
pollution area of each transect.
**2.3.1 Determination of $NO_x$ emission estimates from $NO_2$ measurements**
$NO_x$ emissions were estimated using Equation 4 from the $NO_2$ flux integral and the average $NO_x/NO_2$ ratio ($NO_x$ > 3
ppb) measured by the $NO_x$-analyzer along the route. The emission values were then corrected for expected $NO_x$ loss
during transport using a $NO_x$ lifetime, $\tau$. $NO_x$ emission estimates were calculated as follows:



$$E_{NOx} = E_{NO_2} * \overline{\frac{NO_x}{NO_2}} * e^{\left(\frac{y/w}{\tau}\right)}$$

(4)

where $\tau$ is $NO_x$ lifetime, w is wind-speed, and y is the distance between the $NO_x$ source and the measurement
location. For routes where individual $NO_x/NO_2$ ratios deviated significantly from the route average, the $NO_x$
emission estimates were calculated by applying 1) the route-averaged $NO_x/NO_2$ ratio and 2) individual $NO_x/NO_2$
ratios associated with each $NO_2$ VCD point by point. Multiple factors determine $NO_x$ lifetime in a plume. A $NO_x$
lifetime of 6 hours was used in this study based on considerations given in section 3.3. A sensitivity analysis was
performed varying the lifetimes between 4-8 hours (Supplement S7). The conversion factors used to calculate $NO_x$
emissions for each route can be found in Table S8. The $NO_x/NO_2$ ratios are more fully addressed in Section 3.2 and
the $NO_x$ lifetime is addressed in Section 3.3.
**3 Results & Discussion**
**3.1 Relationship between $NO_2$ VCDs & $NO$-$NO_2$-$NO_x$ Analyzer Measurements**
Figure 2 shows that enhancements in $NO_2$ VCDs downwind of Sarnia were generally associated with $NO_2$ surface
mixing ratios enhancements during Days 1 and 2. This suggests that pollution from Sarnia was well-mixed within
the boundary layer at the measurement locations, typically 14-23 km downwind of sources (Figs. 3 & 4). However,
the ratio of $NO_2$ VCD to $NO_2$ mixing ratio was sometimes variable even during relatively short time periods when
the boundary layer height was expected to be constant (Fig. 2a). This variability was probably due to the presence of
multiple $NO_x$ plumes that had originated from sources with different heights (i.e., stacks and surface sources) and
emission rates.
In contrast to Days 1 and 2, $NO_2$ VCD enhancements on Day 3 were not consistently associated with $NO_2$ surface
mixing ratio enhancements (Figs. 5 & 6). A large surface enhancement ($NO_x$=22 ppb) was observed at the location
of the VCD $NO_2$ enhancements ($\sim 2.5 \times 10^{16}$ molec cm$^{-2}$) associated with the NOVA Chemicals industrial plume on
route 2 (Figs. 5b & 6b) but not on route 1 (Fig. 5a & 6a). This discrepancy is likely due to the closer proximity of
the driving route to the source compared with Day 1, combined with limited vertical mixing of the plume. The
relatively long sampling time of the $NO_x$ analyzer with a relatively fast driving speed on this route may also have led
to an underestimation of the true $NO_x$ values for this localized plume.





### 3.2 $NO_x/NO_2$ Ratios

The $NO_x/NO_2$ ratio is necessary to estimate $NO_x$ emissions from the source, given measurements of $NO_2$ VCD's (Eq. 4). Ratios of $NO_x/NO_2$ (Table 2) measured along the routes on Days 1 and 3 were within 20% of the route-averaged value with a relative standard deviation of less than 12%. $NO_x/NO_2$ ratios tended to increase at locations associated with transported plumes' centerlines, as expected due to an increase in NO emissions from the sources (see Fig. 7), and exhibited the greatest variability in air-masses affected by sources with different altitudes and emission rates. Day 1, route 1 exhibited variable $NO_x/NO_2$ ratios due to emissions from the power-plants across the river in Michigan, residential and vehicular traffic, and industrial emissions (Figs. 3a & 7).

Potential errors may exist in the $NO_x/NO_2$ ratio due to the presence of other $NO_z$ species in the air mass (e.g., $HNO_3$, HONO, $NO_3$, $N_2O_5$, organic nitrates, etc.) that are also converted to NO by the Mo-convertor in addition to $NO_2$ (Dunlea et al., 2007). However, these errors are smaller than might be expected due to the presence of the error in both the numerator and the denominator of the $NO_x/NO_2$ ratio, thus offsetting. For example, at an apparent $NO_x/NO_2$ ratio of 1.40 (average in Table 2), a 10% and 30% error in the reported $NO_2$ due the presence of other $NO_z$ species gives rise to errors of only -2.6% and -6.6% in the measured $NO_x/NO_2$ ratio respectively. Mathematically, the error in the $NO_x/NO_2$ ratio gets larger as the percentage of NO in the total $NO_x$ increases. However, since most of the interfering $NO_z$ species are generated photochemically, or only at night ($NO_3$, $N_2O_5$) increasing with reaction time and distance away from the source, the percentage of interfering species is smaller at higher values of total NO and $NO_x$. Under significantly intense photochemical conditions in the MCMA-2003 field campaign in Mexico, the interference in the chemiluminescence monitors resulted in average $NO_2$ concentrations being 22% higher than those determined from spectroscopic measurements (Dunlea et al., 2007), which would give rise to an error in the $NO_x/NO_2$ ratio of <5%. In the current study we estimate that the resultant negative bias in the measured $NO_x/NO_2$ ratio does not exceed -5% for several reasons; i) we filter out low $NO_x$ data (<3ppb), ii) the emission integral is dominated by regions with high $NO_x$ that are spatially and temporally close to the sources and, iii) photochemistry was reduced during this spring campaign. The uncertainty that arises from potential errors in the $NO_x/NO_2$ ratio is insignificant compared to other errors (see Supplemental Table S9). It is also worth noting that $NO_2$ measurements by the $NO_x$ analyzer are not directly used for the calculation of emissions; only the $NO_x/NO_2$ ratio is used.


Previous Mobile-MAX-DOAS studies have relied on literature estimates of the $NO_x/NO_2$ ratio (Shabbir et al., 2016;
Shaiganfar et al., 2011) or estimated the ratio from a Leighton ratio calculated using local air quality station data
(Ibrahim et al., 2010). In regions with many pollutant sources throughout (e.g., megacities), this ratio is expected to
be horizontally and vertically inhomogeneous. The ratio can therefore be challenging to estimate and can increase
the uncertainty of the $NO_x$ emission estimate. Estimation of $NO_x/NO_2$ ratios from near-surface monitoring stations
can be problematic because the ratios are applied to a VCD but may reflect only local emissions (e.g., nearby
vehicular exhaust) rather than the total boundary layer. In this study, $NO_x$ data impacted by local emissions were
removed. Also, the Sarnia emissions were expected to be well mixed to the surface since most of the transects were
driven sufficiently far from the sources. Therefore, the near-surface $NO_x/NO_2$ ratios should be representative for the
altitude range of the dispersed $NO_x$ plume(s). This hypothesis is supported by the similarity between the $NO_2$
surface and VCD temporal trends during the study, especially on Days 1 and 2 (Fig. 2).
**3.3 $NO_x$ Lifetime**
Various lifetimes of $NO_x$, $\tau$, have been used in previous mobile MAX-DOAS studies for the calculation of $NO_x$
emissions from $NO_2$ measurements: 6 hr in Germany (Ibrahim et al., 2010), 5 hr in Delhi (Shaiganfar, 2011), 5 hr in
China (Wu et al., 2017) and 3 hr summer – 12 hr winter in Paris (Shaiganfar, 2017). In Beirle et al. (2011), the
daytime lifetime of $NO_x$ was quantified by analyzing the downwind patterns of $NO_2$ measured by satellite
instruments and shown to vary from ~4 hr in low to mid-latitude locations (e.g., Riyadh, Saudia Arabia) to ~8hr in
northern locations in wintertime (e.g., Moscow, Russia). In a follow up study, Valin et al (2013) showed that one
cannot assume that $\tau$ is independent of wind speed and derived values of $\tau$ from the satellite observations over
Riyadh to be 5.5hr to 8 hr, corresponding to OH levels of $5\text{-}8\times10^6$ molec $cm^{-3}$ at high and low wind speeds.
Multiple factors determine $NO_x$ lifetime in a plume, including season (e.g., insolation) (Liu et al., 2016), latitude,
wind-driven dilution (Nunnermacker et al., 2000; Valin et al., 2013), $NO_x$ emission rate and initial dilution
(Nunnermacker et al., 2000), temperature, hydroxyl radical levels (OH) and precursors to OH including $O_3$, $H_2O$,
and HONO. Very importantly, the daytime lifetime of $NO_x$ is a nonlinear function of the $NO_x$ concentration itself,
having longer lifetimes at high and low concentrations with the shortest lifetimes at intermediate $NO_x$ concentrations
due the impact on OH levels in a non-linear feedback on its own lifetime (Valin et al., 2013). The $NO_x$ lifetime is
ultimately dependent on the OH levels since this dictates the loss rate of $NO_2$ to its terminal sink ($NO_2$ + OH ➜
HNO$_3$).  However the presence of VOC's in the urban plume, which are catalytically oxidized forming O$_3$ in the
presence of NO$_x$ and HO$_x$ (OH + HO$_2$), can decrease the NO$_x$ lifetime due to their acceleration of the conversion of
NO to NO$_2$ via peroxy radical reactions (RO$_2^.$ + NO → NO$_2$ + RO$^.$).  Therefore, NO$_x$ lifetimes can vary both
spatially and temporally (Liu et al., 2016), even within the same plume (Valin et al., 2013). Underestimation of the
true NO$_x$ lifetime leads to overestimation of the NO$_x$ emissions, while an overestimate leads to underestimation of
the emissions.
While photolysis of HONO is often the major source of OH in the morning boundary layer (Platt et al., 1980; Alicke
et al., 2002), midday production of OH via photolysis of O$_3$ and subsequent reaction of O ($^1$D) with water is
frequently the dominant source of OH.  Assuming O($^1$D) is in steady-state, it can be shown that when ozone
photolysis is the main source of OH, the product of the mixing ratios of H$_2$O and O$_3$ is proportional to the
production rate of OH. In this study, the [H$_2$O]*[O$_3$] product was calculated using surrounding station measurements
(see Supplement S8.1). The [H$_2$O]*[O$_3$] product indicates that mid-day OH production under the spring-conditions
for Days 1 and 2 is only 10-25% of the expected OH production under warmer more humid summer-conditions,
presuming that O$_3$ photolysis predominates. This might suggest OH levels were lower in our study than during
summer, and hence NO$_x$ lifetimes longer, however we assume this with caution as the HONO production is not
known nor are the loss rates of OH.
As mentioned, the presence of VOC's can decrease the lifetime of NOx under conditions where NOx  is sufficiently
high to dominate the peroxy radical reaction path.  To test for the presence of VOC's in the plumes (in the absence
of measurements), Leighton ratios, $\phi$  (Leighton, 1961), were calculated at locations of maximum NO$_2$ VCD
associated with Sarnia plumes. Leighton ratios were calculated following Eq. (5) (see Supplement S8.2 for details):

$$\phi = \frac{j_{NO2}[NO_2]}{k_8[NO][O_3]} \qquad (5)$$

where j$_{NO2}$ is the NO$_2$ photolysis rate, k$_8$ is the temperature-dependent rate constant for the reaction between NO and
O$_3$. Leighton ratios equal to 1.0 indicate that NO, NO$_2$ and O$_3$ are in steady state with no significant interference
from other species, while ratios of $\phi$ greater than 1.0 imply the role of other peroxy radical species (e.g., RO$_2$, HO$_2$)
in the conversion of NO to NO$_2$ (Pitts and Finlayson-Pitts, 2000). The NO$_2$/NO ratios were obtained from the NO$_x$
analyzer measurements, O$_3$ mixing ratios were obtained from local monitoring stations during the same daytime



periods as the transects. Values of $j_{NO2}$ were estimated using SLEA Moore Line station solar irradiance data (Fig. 1;
Table S1) and solar zenith angle following the method in Wiegand and Bofinger (2000).
Table 2 shows Leighton ratios calculated at the locations of maximum $NO_2$ VCD enhancements. Calculated
Leighton ratios were significantly greater than 1 (1.7-2.3) at peak $NO_x$ locations on Day 1 (Table 2). We interpret
this as an indication that significant levels of peroxy radicals were present in the plume, presumably from VOC
oxidation by the OH radical.  This is consistent with high VOC emissions from the petrochemical facilities in
Sarnia, with emission rates >300 tonnes yr$^{-1}$ each for four of the top six industrial $NO_x$ emitters in Sarnia
(Environment and Climate Change Canada, 2018d). The Day 2 Leighton ratio of less than 1.0 in Table 2 suggests a
relatively fresh plume (only 4 km downwind of a facility) that had not come to photo-stationary state.
Thus we have indications that OH production may be lower than summer time leading to longer NOx lifetimes and
we have indications that VOC oxidation in the plume may be significant leading to shorter NOx lifetimes than air
masses where the photo-stationary state in $NO_x$ is valid.  Without further information, we have opted to assume a
central $NO_x$ lifetime assume of ~ 6 hr. Sensitivity calculations were conducted for $NO_x$ emission estimates using a
range of lifetimes of 4-8 hours (Supplement S7). Varying the lifetime from ± 2 hours changed the emission estimates
by <15% for all routes except for Day 1 route 1 due to low wind-speeds during that route (30% change).
For the calculation of $SO_2$ emissions, $SO_2$ was assumed to have a sufficiently long lifetime in the boundary layer so
as to be conserved between the emission and measurement location. Note that cloud processing of $SO_2$ was assumed
to be negligible since $SO_2$ measurements were completed on a mostly cloud-free day.
**3.4 Emission Estimates**

**3.4.1 Emission Estimates of Sarnia**
The VCDs measured are shown in Fig. 3-6 while the $NO_x$ emissions calculated using Eqs. (3) and (4) are shown in
Table 4. The values of $VCD_{influx}$ required for the calculations were typically determined from measurements of VCD
in low pollution transect areas. However, the $VCD_{influx}$ on Day 2 was not determined in this way since these DSCDs
were close to zero within error (Figs. 2 & 4). The $VCD_{influx}$ is expected to be low on Day 2 because the north wind-
direction indicates that the air-masses originated from over Lake Huron. These low values were probably due to low





light levels during measurement, insufficiently long integration times (low signal to noise ratio) and $NO_2$
background VCD values below the instrument's limit of detection. A low value of $VCD_{influx} = 0.5(\pm 0.5) \times 10^{15}$ molec
$cm^{-2}$ was therefore assumed.
The emissions were calculated in two ways i) using a route-average $NO_x/NO_2$ ratio value for each route estimate and
ii) using individual $NO_x/NO_2$ ratios co-located with each VCD measurement. For Day 1 route, the route average
$NO_x/NO_2$ ratio was $1.53 \pm 0.12$ ppb $ppb^{-1}$ with the difference between the calculated emission rates using the two
methods being only 3%. Day 1 transects 2-4 exhibited small variability in $NO_x/NO_2$ (Table 4) and the variation in
the $NO_x/NO_2$ ratio impacted emission estimates by less than 5%.
However, the difference between emission estimates calculated using individual $NO_x/NO_2$ ratios versus a route-
averaged value can be non-trivial, as observed with the Day 2 route 1. Day 2 had consistent northerly wind
conditions, and east-west transects were driven south of Sarnia to capture the urban plume and background regions
to the east (Fig. 4). The resultant Sarnia $NO_x$ emission using the first method is consistent with the first three Day 1
emission estimates but application of the second method (individual $NO_x/NO_2$ ratios collocated with each VCD)
increased the emission estimate by ~50% (Table 4 and Fig. 8). The $NO_x/NO_2$ ratio was generally consistent with the
averaged value of 1.3 (maximum $NO_x/NO_2$ removed) but increased to 3 in the region of maximum $NO_2$ VCD
enhancements 7 km south of the NOVA Chemicals facility (Table 3). The calculated Leighton ratio for this peak
$NO_x/NO_2$ ratio location is less than 1 (see 3.4.2 and Table 3). The Leighton ratio suggests the plume from the
NOVA Chemical facility had significant NO that had not had sufficient time to come to a photostationary state. The
emission estimate using individual $NO_x/NO_2$ ratios is considered the more accurate value for this route compared to
the emission value calculated using the route-averaged ratio.
The importance of measuring the local $NO_x/NO_2$ ratio is also illustrated by observing variation of the ratio due to the
impact of the Michigan power-plants' plume, apparent in the Day 1 route 1 East-West transect (Fig. 3a). The
$NO_x/NO_2$ ratio along this transect increased to ~1.7 (Fig. 7), higher than the maximum $NO_x/NO_2$ ratio observed in
the North-South transect downwind of Sarnia. A higher ratio is somewhat unexpected because the distance between
the source and receptor measurement for the power plant source was greater than the source-receptor distance for the
Sarnia sources. Thus, the power-plant plume would have been expected to be more aged, but the results suggest that
the power-plants' plumes had a slower conversion of NO to $NO_2$ perhaps due to higher initial mixing ratios of $NO_x$





(Nunnermacker et al., 2000). Very high NO mixing ratios in a power plant plume (i.e., > 40ppb) could completely
titrate the ambient $O_3$ in the air entrained into the plume, an observation previously seen in power plant plumes
(Brown et al., 2012).
The East-West transect appears to have captured approximately half of the power-plants' plume since the $NO_2$
VCDs and the $NO_2$ mixing ratios increase from background to a plateau at a maximum (Fig. 2a). A preliminary
estimation of the $NO_x$ and $SO_2$ emissions from the power-plants can be determined by doubling the flux integral
calculated from this East-West transect. To do this, we have used $VCD_{influx} = 2\text{-}3\times10^{15}$ molec cm$^{-2}$ for $NO_x$ and zero
for $SO_2$ since the background region $SO_2$ DSCDs were at or below detection limits. The $NO_x$ estimate used
individual $NO_x/NO_2$ ratios because the $NO_x/NO_2$ ratio was significantly higher in the plume than outside the plume.
This illustrates the importance of in-situ instruments of $NO_x/NO_2$, especially when close to the source where plume
$NO_x/NO_2$ ratios can be variable (Valin et al., 2013). Given the above assumptions, a tentative estimate of the total
emissions from the power plants are 0.31-0.46 tonnes $NO_x$ hr$^{-1}$ and 0.77 tonnes $SO_2$ hr$^{-1}$, respectively. The hourly
emissions of the power-plants downscaled from reported 2015 annual values are 0.74 tonnes $NO_x$ hr$^{-1}$ and 2.56
tonnes $SO_2$ hr$^{-1}$ (United States EPA, 2018). Our hourly estimates are only preliminary since only half of the plume
(approximately) was captured by the measurement transect.
The $NO_x$ emission estimates from Sarnia from Day 1 are consistent within 25% and are consistent with the Day 2
estimates within the calculated error of approximately ±45% (Fig. 8, Table 4). Some variability between the
emission estimates is expected due to wind-data uncertainties, $NO_x/NO_2$ vertical profile variability, errors introduced
by using a constant $VCD_{influx}$ and $NO_x$ lifetime, and temporal variations in emissions from the source.
Conversion of the hourly measured emissions to annual emissions would require knowledge and application of
daily, weekly and seasonal emission profiles, which is beyond the scope of this work. The Mobile-MAX-DOAS
emission estimates are reported in units of tonnes per hour since routes were completed within <40 minutes. Events
such as flaring can significantly increase the instantaneous emission rate but are excluded from the annual emission
inventory data. However, there was no reported flaring during the measurement period (MOECC 2017; personal
communication). $NO_x$ emissions from petrochemical facilities, excluding flaring, typically have low variability
during periods of continuous operation. According to Ryerson et al. (2003), variation in average hourly $NO_x$
emissions from a petrochemical facility reported by industry (CEMS data) was <10% from an average of the hourly





average emissions over 11 days in Houston, Texas. However, this trend may be different for the chemical industry.
A first-order comparison to the 2017 National Pollution Release Inventory (NPRI) values (downscaled by assuming
constant emissions) was made to determine whether our measured Sarnia emissions are reasonable. The NPRI value
is the sum of the $NO_x$ emissions from the top 9 industrial emitters of $NO_x$ in Sarnia whose emissions would have
been captured along the driving routes. The NPRI requires significant point source industry facilities to report their
pollutant emissions, but the method of estimating emissions can vary by facility (Canada and Canada, 2015). The
NPRI emission value does not include mobile and area sources from the Sarnia region. Thus, the NPRI emission
inventory values for Sarnia would be expected to be smaller than our measured emissions because of this exclusion.
The measured $NO_x$ emissions are larger than the 2017 NPRI value but not statistically so (Fig. 8; Table 4). The
exception is the Day 1 route 1* value, which is statistically higher. The average of the four $NO_x$ emission estimates
from Sarnia is greater than the 2017 NPRI value. These results demonstrate that our measured emission rates are
reasonable. Future Mobile-MAX-DOAS studies could focus on determining diurnal trends in emissions by driving
multiple routes at as many times of the day as possible on multiple days, seasons and weekdays/weekends.
Measurements of vertical wind profiles could reduce emission uncertainty to allow identification of temporal trends
by comparing same-day measurements.
Apart from $NO_x$, we were also able to estimate $SO_2$ emissions from the Sarnia urban/industrial region from the Day
1 route 3 (Table 5). Our $SO_2$ emission estimate using the 10 m wind-speed is consistent within error with the 2017
NPRI value (Table 5). We expect our $SO_2$ emission estimate to be closer to the NPRI values compared to the $NO_x$
estimates because $SO_2$ emissions from area and mobile sources in Sarnia are expected to be small relative to
industrial sources (Ministry of the Environment and Climate Change, 2016). Since ships were not operating in the
St. Clair River at this time of year, shipping emissions of $SO_2$ were absent. Thus $SO_2$ plumes in this region are
localized to the major industrial emissions sources. Therefore, the VCDs from the areas unaffected by the Sarnia
plumes are representative of background values, $VCD_{influx}$. While the Mobile-MAX-DOAS was able to capture these
plumes (Fig. 9), only 1 of 7 local monitoring stations (LaSalle Road, Fig. S1) observed elevated levels of $SO_2$ during
this period. The under-sampling by stations is due to the highly localized nature of the $SO_2$ plumes that are from
stacks where the plume is frequently elevated above the surface. These results illustrate the complementary nature of
Mobile-MAX-DOAS and in-situ measurements and the importance of monitoring techniques that can capture
localized plumes independent of the wind direction.


### 3.4.2 Emission Estimates of NOVA Chemicals Industrial Facility


$NO_x$ emissions were opportunistically measured from a single facility on Day 3 because the southerly wind-
directions isolated this plume (Environment and Climate Change Canada, 2018b) from other industrial sources in
Sarnia. The plume originated from Nova Chemicals, the $2^{nd}$ highest emitter of $NO_x$ in the region in 2017. These
conditions allowed us to test the mobile-MAX-DOAS method in isolating a single plume. The wind-direction on
Day 3 indicated that the air-masses originated from rural areas south of Sarnia and the $VCD_{influx}$ was expected to be
low, ~ $1\times10^{15}$ molec $cm^{-2}$.
The emission estimates of $NO_x$ from the two routes on Day 3 from the NOVA Chemicals industrial site (Tables 4 &
5) are consistent with each other within 10%. The consistency increases confidence in fitting the spectra in each
transect against a local FRS and removing influx using the average "background" VCDs rather than using the
"$DSCD_{Offset}$" method in this case. The use of "background" VCDs is appropriate because vehicular traffic upwind of
the measurement transect is minimal in the local area. Upwind emissions were unlikely to have contributed
significantly to the total measured emissions. The emission estimates from NOVA Chemicals are larger than the
2017 NPRI value (Tables 4 & 5). This comparison merely indicates that the Mobile-MAX-DOAS values are
reasonable given that there was likely diurnal variability and the measurements were taken only during a single hour
on a single day.

### 3.5 Comparison of OMI Satellite and MAX-DOAS VCDs


The satellite and MAX-DOAS $NO_2$ VCDs on Day 1 exhibit similar spatial trends in the simple sense that $NO_2$
VCDs increase towards the south from the background regions north of Sarnia (Fig. 10). This trend is probably due
to a combination of emissions from U.S.A. power-plants, the Detroit area as well as Sarnia. The $NO_2$ VCD of the
pixel containing the majority of the Sarnia industrial facilities is comparable to rural area VCDs to the north-west of
Sarnia. Only $1/8^{th}$ of the "Sarnia" pixel's footprint region is likely to be impacted by Sarnia emissions, and the
remainder observes mostly rural to semi-rural regions. The OMI Pixel from Day 3 (Fig. 11) containing Sarnia
exhibits a minimal increase in $NO_2$ VCD ($1\text{-}2\times10^{15}$ molec $cm^{-2}$) compared to the surrounding background regions
(Fig. 11). In contrast, the Mobile-MAX-DOAS measurements observed VCD enhancements of up to $1\times10^{16}$ molec
$cm^{-2}$ within this pixel. The averaging due by the large pixel size (24 km×84 km) causes underestimation of the



maximum VCDs. Identification of Sarnia-only emissions without error due to horizontal averaging or inclusion of
other sources may require satellite measurements with nadir-viewing pixels centred on Sarnia and/or extremely large
averaging times.
**3.6 Uncertainties in this Study and Recommended Improvements for Mobile-MAX-DOAS Measurements**
Many of the factors that increased the uncertainty in the emission values in this study can be significantly reduced in
future through relatively small changes in the method. The many factors have been addressed in Supplemental
Information (section S7) and summarized in Table S9.  Lack of knowledge of the vertical profile of wind-speed
increases uncertainty in Mobile-MAX-DOAS emission estimates since elevated plumes and well-mixed plumes are
transported by winds with typically higher speeds than those near the surface. Future studies could focus on
reducing uncertainty by using measurements from sodar, lidar, tall towers, balloon soundings, or a radio acoustic
meteorological profiler. In this study, uncertainty was increased (18-30% based on sensitivity analysis; see
supplementary S5 & S7) because driving routes could not always include measurements along influx regions (Day
1) due to road proximity to sources or obstructions to the viewing field. Future experiments could measure influx
values while stationary at multiple locations along the upwind region chosen for an unobstructed viewing field. Very
low background trace-gas levels also resulted in zero within error background DSCDs (Fig. 2e). A spectrometer
with a lower limit of detection could solve this issue. Uncertainty in the $NO_x$ lifetime was a small contribution to
uncertainty in this study (up to ±12%) because the distances and transport times between source and measurement
locations were relatively small (<25 km). The exception was Day 1 route 1 where uncertainty was up to 30% due to
low wind-speeds. The error contribution of $NO_x$ lifetime could be non-trivial if driving routes are far from the
sources (e.g., large cities). This error could also be non-trivial if the lifetime that one assumes does not account for
the multiple factors discussed in Section 3.3. Bias in the emission estimates from an incorrect lifetime could be
avoided by determining $NO_x$ lifetimes from photochemical modelling or, for large cities, satellite observations
(Beirle et al., 2011)  but taking into account wind speeds (Valin et al., 2013).
**4 Conclusions**
In this study, we combined Mobile-MAX-DOAS techniques with mobile $NO_x$ measurements and a modular
meteorological station to measure emissions of $NO_x$ and $SO_2$ from the Sarnia region, a relatively small
urban/industrial city. Trace-gas VCDs were determined using the $DSCD_{offset}$ method (Wagner et al., 2010) or by



fitting measured spectra against a route-local low pollution spectrum. Both methods provided good results, which
suggest that the first method is ideal if there are many hours of measurements while the second method is ideal when
short routes contain low-pollution regions. Average lower limit Mobile-MAX-DOAS emissions of $NO_x$ from Sarnia
were measured to be $1.60 \pm 0.34$ tonnes $hr^{-1}$ using 10 m elevation measured wind-speeds. The estimates were larger
than the downscaled 2017 NPRI reported industrial emissions of 0.9 tonnes $hr^{-1}$ (Environment and Climate Change
Canada, 2018b) but the NPRI estimate excludes area and mobile emissions. Our lower limit $SO_2$ emission
measurement for Sarnia was $1.81 \pm 0.83$ tonnes $hr^{-1}$ using 10 m wind-speeds, which is equal within uncertainty to
the 2017 NPRI value of 1.85 tonnes $hr^{-1}$ (Environment and Climate Change Canada, 2018c). Our average lower
limit $NO_x$ emission measurement from the NOVA Chemicals Facility was $0.28 \pm 0.06$ tonnes $hr^{-1}$, the same order of
magnitude as the 2016 NPRI value of 0.14 tonnes $hr^{-1}$ (Environment and Climate Change Canada, 2018a).
Simultaneous measurements of $NO$-$NO_2$-$NO_x$ improved accuracy of $NO_x$ emission estimates when plumes of
varying ages were observed. The $NO_x$ results from Days 1 and 2 suggest that accurate Mobile-MAX-DOAS $NO_x$
emission measurements from routes that observe plumes with differing ages require accurate knowledge of the
localized $NO_x/NO_2$ ratio.
The variability in the ratio of the $NO_2$ VCDs and mixing ratios indicates that surface $NO_2$ mixing ratios cannot be
reliably estimated from $NO_2$ VCDs and boundary layer height alone when pollution is emitted from sources of
varying heights and chemical composition. A $NO_x$-analyzer can be an essential component of Mobile-MAX-DOAS
$NO_2$ measurements. The addition of this instrument allows the method to characterize the boundary layer fully and
accurately estimate $NO_x$ emissions from $NO_2$ measurements when multiple $NO_x$ sources are present and when
transects are sufficiently distant from the sources.
The modular meteorological station improved knowledge of local wind essential to identify time periods of low
temporal variability, ensuring low error due to wind estimation. These time periods would have been difficult to
identify with only hourly average or modelled wind data. Accurate knowledge of the vertical wind profile would
significantly enhance the accuracy of the Mobile-MAX-DOAS emission estimates. Future studies could obtain
vertical wind profiles using sodar, lidar, wind-rass, and radiosonde on a weather balloon or local aircraft soundings.



Mobile-MAX-DOAS measurements identified significant OMI intrapixel inhomogeneity and observed industrial
pollution enhancements that were poorly captured by the in-situ ground stations. These results suggest that Mobile-
MAX-DOAS has clear advantages in similar industrial regions over other remote sensing techniques used for
estimating emissions (e.g., using aircraft or satellite): higher spatial resolution, the potential for multiple emission
estimates per day (i.e., observations of diurnal trends), and much lower operational costs. Mobile-MAX-DOAS is a
"top-down" low-cost solution for validating bottom-up inventories that compliments in-situ monitoring and has
significant utility in smaller regions with significant emissions where satellite applications are limited. Future
Mobile-MAX-DOAS studies in such regions can focus on measuring temporal trends in emissions.

**Author Contributions**

ZD conceived of and organized the field campaign with aid from RM. ZD, SB, AK, WF, CC and RM carried out the
experiments in Sarnia. CM modelled conditions for the satellite retrievals of $NO_2$ in the region of Sarnia, and
provided useful advice. ZD and RM prepared the manuscript, with contributions from all co-authors.

**Acknowledgements**

This study was completed with collaborative support by the Ontario Ministry of the Environment and Climate
Change. Funding for the study was provided by NSERC, CREATE IACPES and the York University Faculty of
Graduate Studies. The corresponding author would like to thank Mr. Barry Duffey at the Ontario Ministry of
Environment and Climate Change for his support at the project start. We also thank Tony Munoz of OME for his
continued support of our research.

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



**Table 1** Daily meteorological conditions, number of routes and time period of routes driven. Wind-speed from
SLEA LaSalle Road; Temperature and Relative Humidity from portable meteorological station Day 1 and Day 2 and
from Moore Line station Day 2.

| Date | Number of Routes Driven | Measurement Local Time Period | Average Wind-speed $(km\,hr^{-1})$ | Prevailing Wind-Direction | Average Temperature $(^{o}C)$ | Average Relative Humidity (%) | Emission Area Measured |
|---|---|---|---|---|---|---|---|
| 3/21/2017 | 4 | 10:26-13:16 | 15 | Westerly | 10 | 50 | City of Sarnia |
| 3/22/2017 | 1 | 17:22-17:41 | 8 | Northerly | -3 | 52 | City of Sarnia |
| 3/23/2017 | 2 | 11:10-11:57 | 15 | Southerly | 1 | 42 | NOVA Chemicals Industries Facility |




**Table 2** $NO_x/NO_2$ ratios for routes driven.

| Date | Day's Route Number | Measurement Local Time Period | Number of Points | Average $\pm 1\sigma$ | Median |
|---|---|---|---|---|---|
| 3/21/2017 | 1 | 10:26-11:06 | 37 | 1.53±0.12 | 1.49 |
| 3/21/2017 | 2 | 11:22-11:45 | 23 | 1.45±0.06 | 1.44 |
| 3/21/2017 | 3 | 12:09-12:28 | 18 | 1.36±0.07 | 1.37 |
| 3/21/2017 | 4 | 12:34-13:16 | 24 | 1.29±0.06 | 1.31 |
| 3/22/2017 | 1 | 17:22-17:41 | 10 | 1.49±0.53 | 1.30 |
| 3/22/2017 | 1 | 17:22-17:41* | 9 | 1.32±0.08 | 1.30 |
| 3/23/2017 | 1 | 11:10-11:19 | 5 | 1.39±0.09 | 1.39 |
| 3/23/2017 | 2 | 11:42-11:57 | 9 | 1.46±0.17 | 1.52 |

The 3/22/2017 17:22-17:41* data had the peak $NO_2$ plume location $NO_x/NO_2$ value removed.



**Table 3** Calculated Leighton Ratios for selected plume maximums on Day 1 and 2.

| Date | Local Time | $J_{NO2}$ ($\times 10^{-3}$ s$^{-1}$) | Solar Irradiance (W m$^{-2}$) | Solar Zenith Angle | $O_3$ mixing ratio (ppb) | Measured NO$_2$/NO (ppb ppb$^{-1}$) | Calculated Leighton Ratio |
|---|---|---|---|---|---|---|---|
| 21/03/2017 | 11:00 | 5.23 | 564 | 35 | 18 | 1.7 | 1.61 |
| 21/03/2017 | 11:30 | 5.65 | 600 | 40 | 23 | 2.2 | 1.76 |
| 21/03/2017 | 12:15 | 6.44 | 675 | 43 | 23 | 2.2 | 2.01 |
| 22/03/2017 | 17:28 | 2.71 | 300 | 23 | 10 | 0.5 | 0.44 |







**Table 4** Lower limit $NO_x$ Emission Estimates from 10 m elevation wind-speeds.

| Date | Emission Source | Daily Route Number | Lower-limit $NO_x$ (tonnes hr$^{-1}$) | NPRI $NO_x$ (tonnes hr$^{-1}$) |
|---|---|---|---|---|
| 21/03/2017 | Sarnia | 1 | 1.6±0.8 | 0.9 |
| 21/03/2017 | Sarnia | 2 | 1.2±0.5 | 0.9 |
| 21/03/2017 | Sarnia | 3 | 1.4±0.5 | 0.9 |
| 22/03/2017 | Sarnia | 1 | 1.5±0.6 | 0.9 |
| 22/03/2017 | Sarnia | 1* | 2.2±0.8 | 0.9 |
| 23/03/2017 | NovaChem | 1 | 0.27±0.1 | 0.14 |
| 23/03/2017 | NovaChem | 2 | 0.29±0.1 | 0.14 |

* calculated using individual $NO_x/NO_2$ ratios.





**Table 5** Average NO$_x$ emission estimates from Mobile MAX_DOAS using 10 m wind-speeds and from NPRI.

| | Gas | Lower Limit Emission Estimate (tonnes hr$^{-1}$) | 2017 NPRI Value (tonnes hr$^{-1}$) |
|---|---|---|---|
| Sarnia | NO$_x$ | 1.60±0.34 | 0.9 |
| Sarnia | SO$_2$ | 1.81±0.83 | 1.85 |
| NOVA Chemicals-Corunna Site | NO$_x$ | 0.28±0.06 | 0.14 |




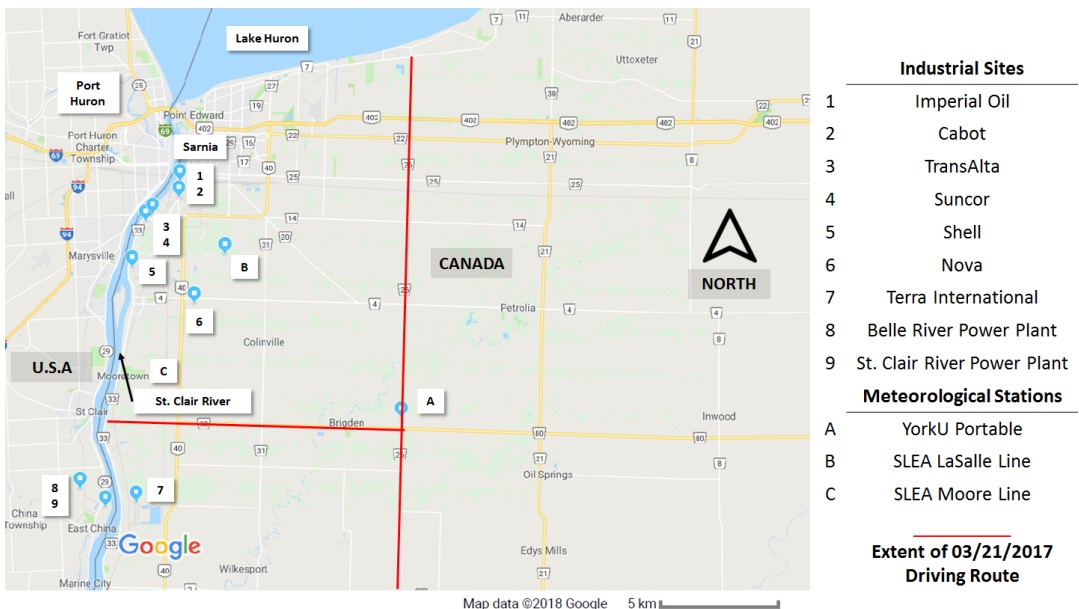

**Figure 1** Location of industrial NO$_x$ and SO$_2$ emission sources and meteorological stations in the Sarnia area.

**Figure 2** NO$_2$ mixing ratios and NO$_2$ VCDs along routes 1-4 on Day 1 (a) – (d)  and route 1 on Day 2 (e).








**Figure 3** Day 1 driving routes; (a) route 1, (b) route 2 and (c) route 3, used to estimate NO$_x$ emissions from Sarnia.






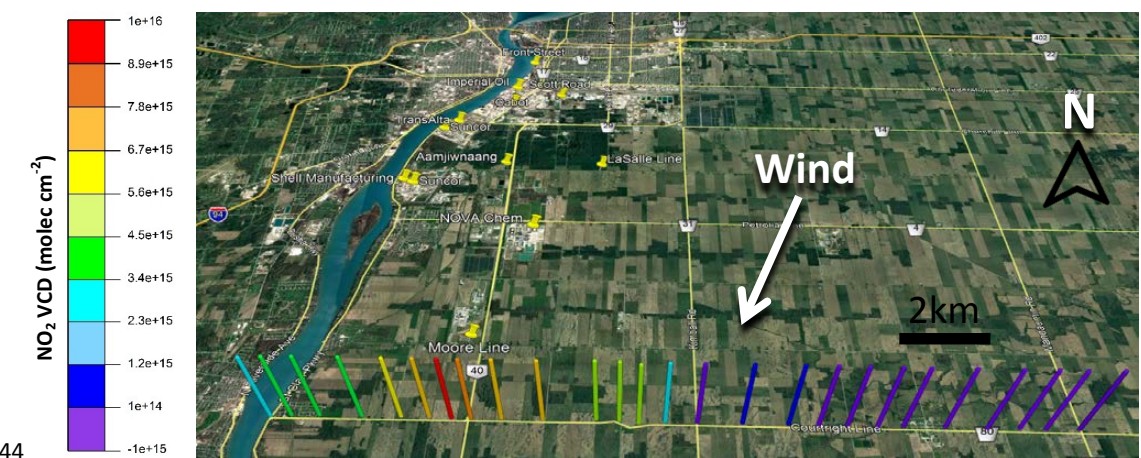



**Figure 4** NO$_2$ VCDs measured on Day 2 route 1.




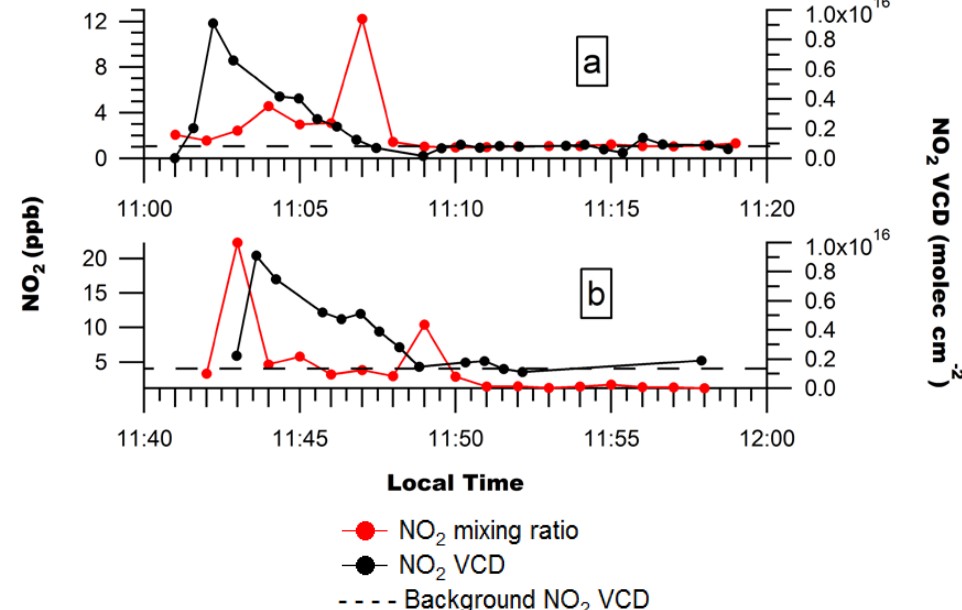



**Figure 5** $NO_2$ mixing ratios and $NO_2$ VCDs measured on Day 3 along (a) driving route 1 and (b) driving route 2.



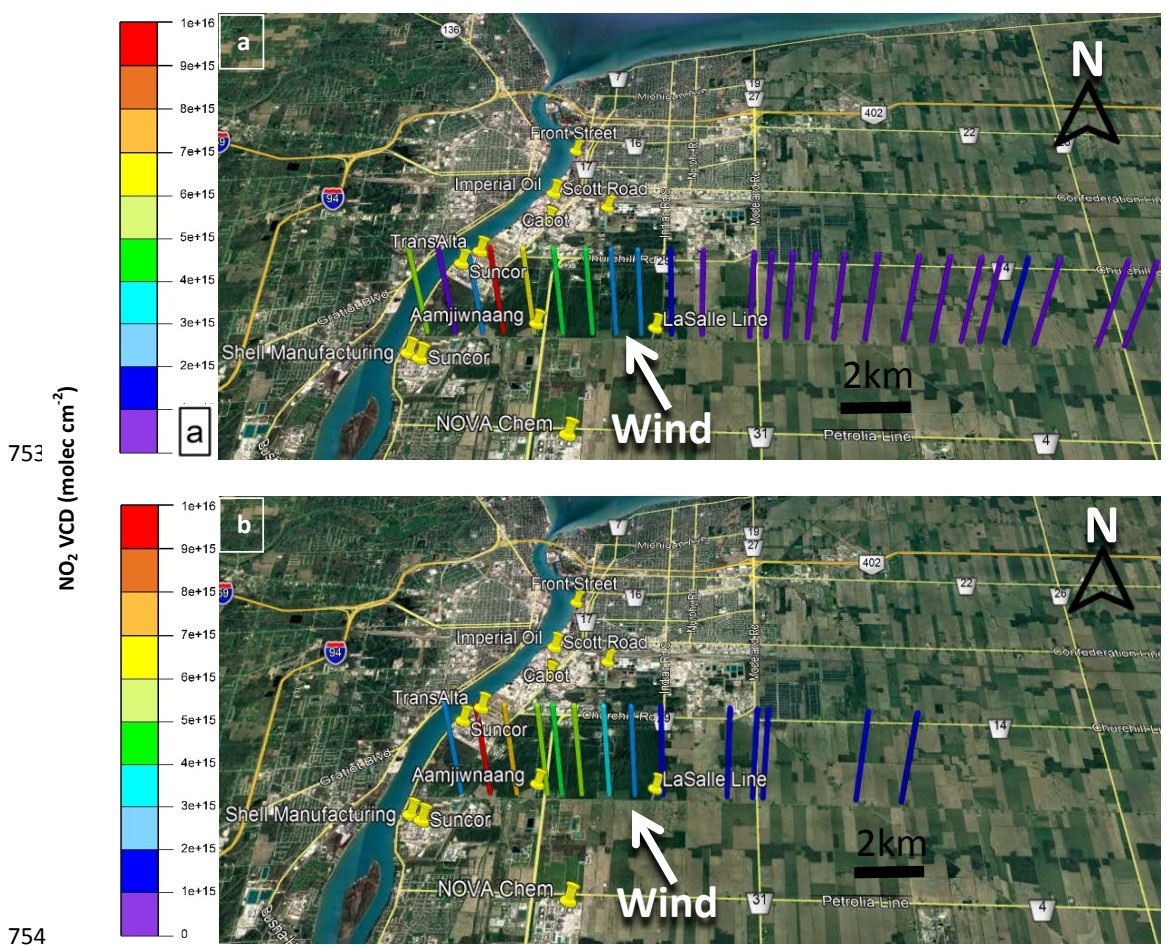



**Figure 6** NO$_2$ VCDs measured on Day 3 along (a) driving route 1 and (b) driving route 2.







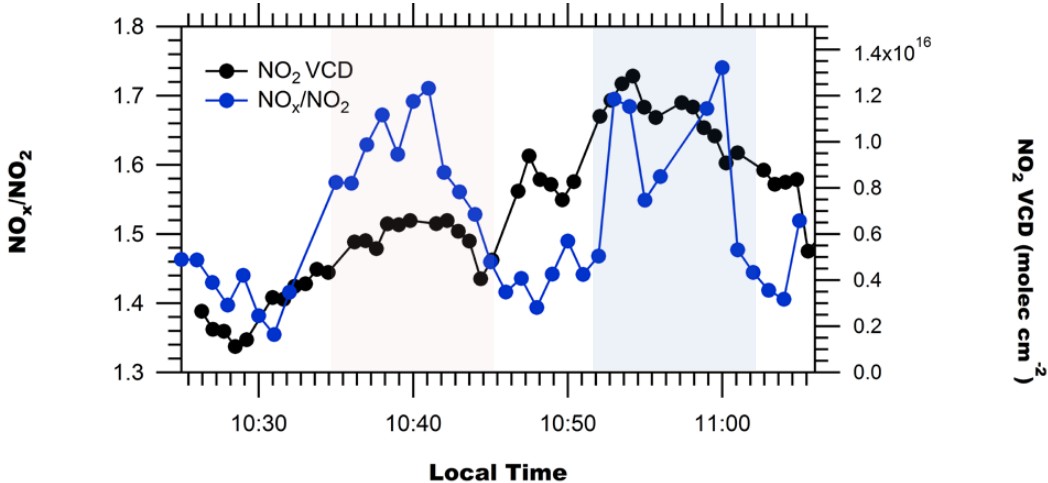

**Figure 7** NO$_2$ VCDs and NO$_x$/NO$_2$ ratios on Day 1 route 1. Detection of Michigan power plants' plume(s) (left) on

East-West transect & Sarnia plume (right) on North-South transect are highlighted in pink and blue, respectively.





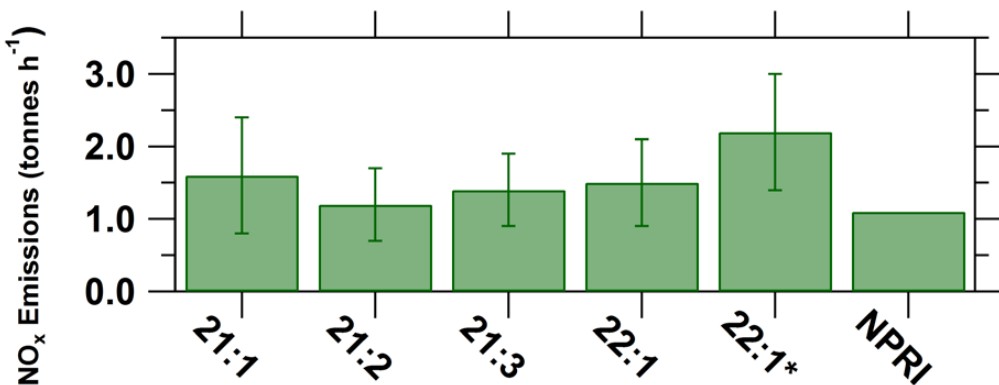


**Figure 8** Lower limit estimates of $NO_x$ Emissions from Sarnia on Day 1 and Day 3 and 2016 NPRI emissions. The

22:1* $NO_x$ emission estimate used individual $NO_x/NO_2$ ratio values for each VCDs rather than a single average

ratio.








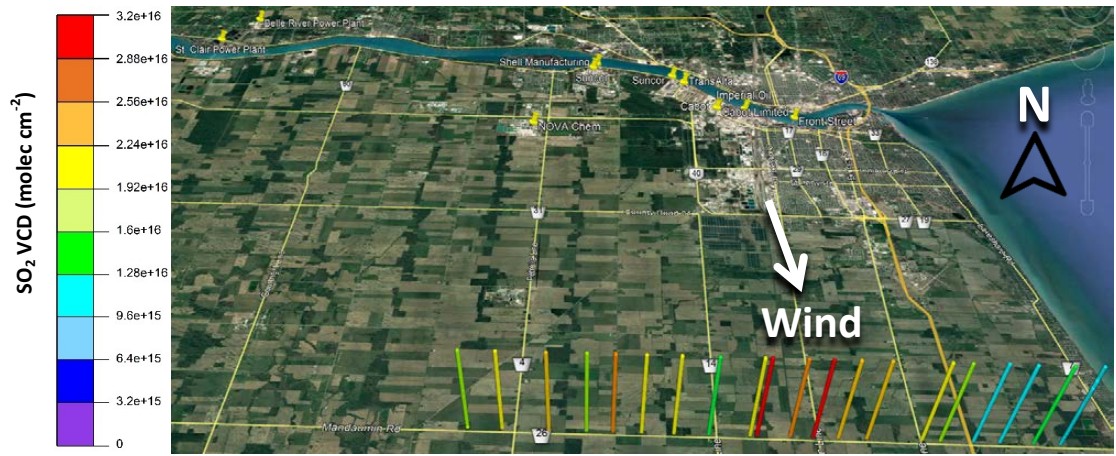


**Figure 9** SO$_2$ VCDs along route for emission estimate (Day 1 Route 3).

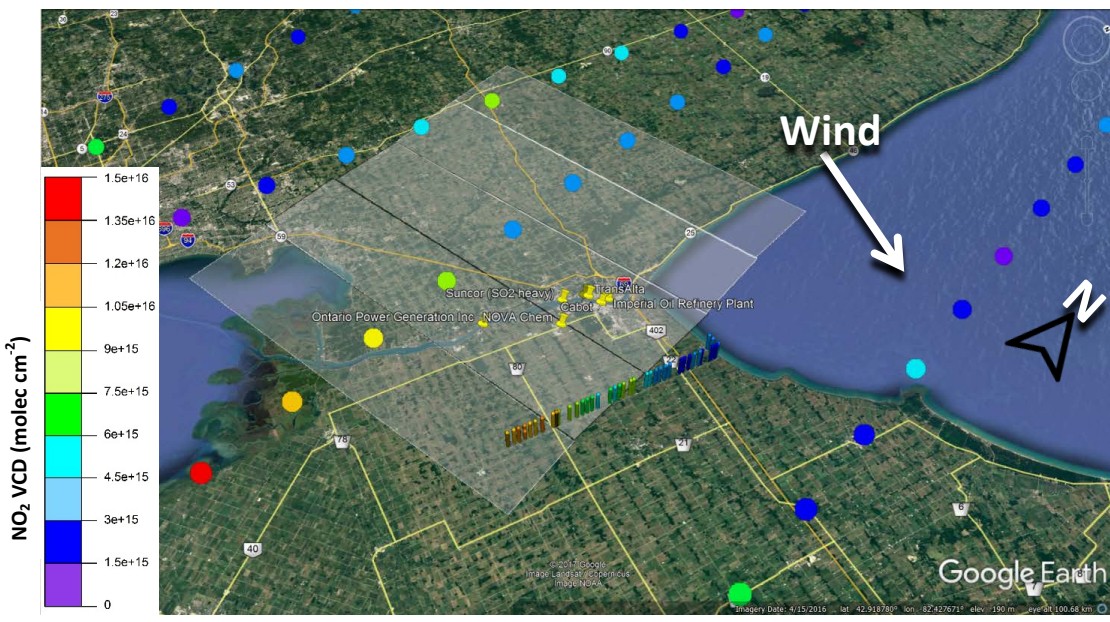


**Figure 10** Day 1 NO$_2$ VCDs from OMI satellite VCDs and mobile-MAX-DOAS Route 4. OMI satellite pixels
closest to Sarnia were measured at ~18:00 local time. Semi-opaque rectangles centered on the colored dots
(indicating satellite VCD value) indicate the spatial extent of the pixel.




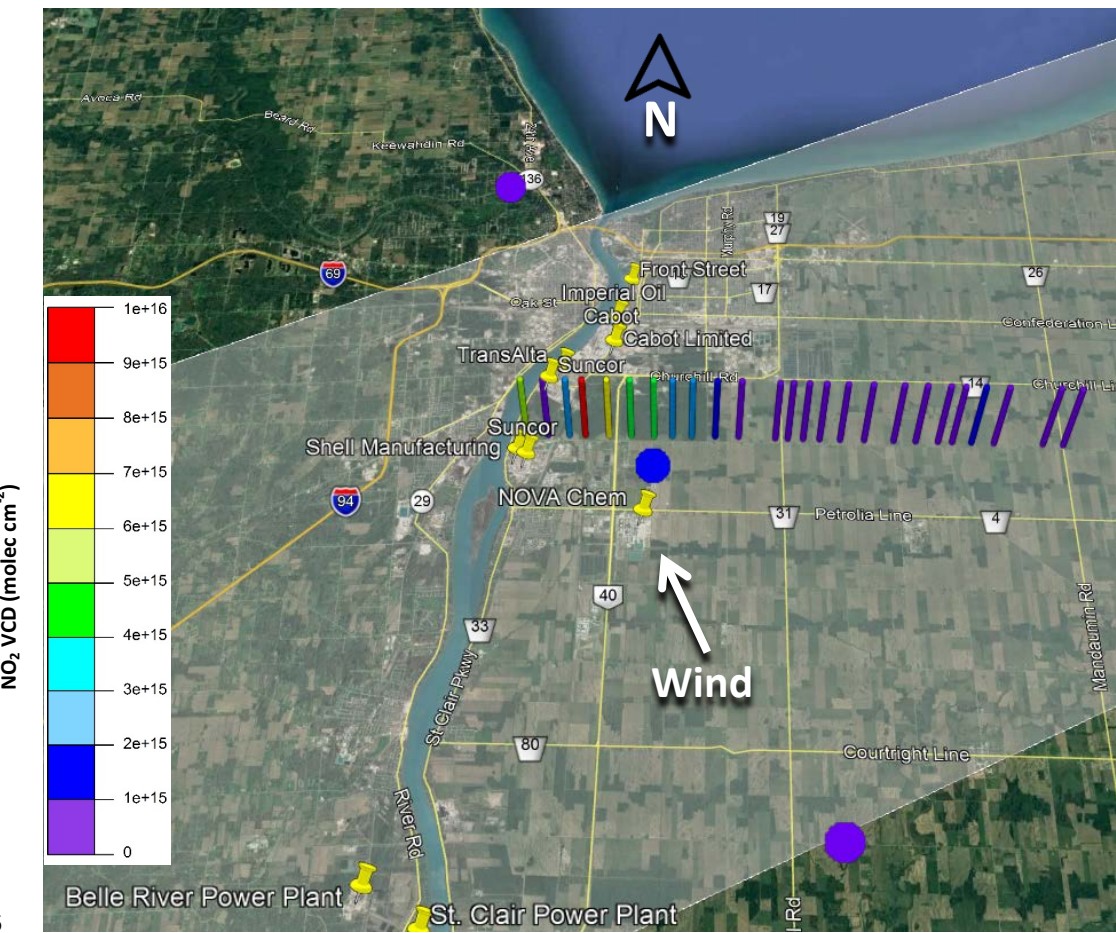


**Figure 11** Day 3 NO$_2$ VCDs from OMI satellite and mobile-MAX-DOAS Route 1. OMI pixels shown were

measured at ~18:00 local time. Semi-opaque rectangle centered on the colored dots indicates the spatial extent of the

pixel.