# Peer review of "Estimation of NOx and SO2 Emissions from Sarnia, Ontario 1 using Mobile-MAX-DOAS and a NOx-Analyzer 2"

_Atmospheric Chemistry and Physics, 2019_

## Referee Comment (RC1) · Anonymous Referee #1 · 19 Aug 2019

The manuscript by Davis et al. describes mobile multi-axis DOAS observations of industrial and urban emissions around the town of Sarnia, ON. The study is based on 3 days of observations of UV-vis absorption spectra in 30, 40, and 90 degree elevation viewing angles from a moving car downwind, and sometimes upwind, of the emission sources. The spectra were analyzed to retrieve $NO_2$ and $SO_2$ column densities. These were then converted into fluxes using 10m wind data. In addition, an in-situ NOx monitor installed on the vehicle was used to convert $NO_2$ to NOx columns. The use of the NOx monitor is a nice touch as it reduces one of the main uncertainties when using DOAS for NOx flux measurements. The manuscript is thorough in its discussion of the methodology, and the authors should be commended for the detailed discussion of

the uncertainties of their observations. The authors provide a preliminary comparison of their fluxes with those from a 2015 power plant emission inventory and a 2017 industrial facility emission inventory. The comparison is reasonable, considering the various uncertainties entering the determination of both emission rates.

Overall, this is a very good manuscript, although I am wondering if it would have been better suited for Atmospheric Measurement Techniques, since most of the manuscript is dedicated to explaining the use of the MAX-DOAS technique to measure emissions. Maybe it would be worth discussing the emission results in more detail, i.e. a comparison with some other studies that show whether the agreement with the emission inventory is similar to other locations and/or addressing the overall question of the accuracy of emission inventories. This would make this study more valuable for ACP readers. Aside from these general comments I have a few other minor comments that should be addressed before the manuscript can be published in ACP.

- I am a little confused about the $SO_2$ emission estimates. It seems those are only reported for day 1. What about the other days? Since $NO_2$ data is available I assume that $SO_2$ is available as well. If the $SO_2$ was not above the detection limit it should be reported as upper limit emission estimate.

- Section 2.2: Was HCHO included in the fit of $SO_2$? And more generally, why was HCHO not retrieved? HCHO column densities would provided information on VOC's that are discussed later in the manuscript.

- Section 3.4.1: Are the emissions used for the NPRI comparison also scaled up to account for not measuring the entire plume?

- Lines 270-271: "…thus offsetting."? I do not understand this sentence.

- Lines 478 – 480, and other locations in the manuscript: Please clarify that one needs both the vertical wind profile, as well as the trace gas profile, for accurate flux determination. Wind profiles alone, while improving the calculation, are insufficient. Have you considered the change of wind direction in the boundary layer (Ekman spiral)?

- Figures 2, 5, and 7: Add error bars, or at least list the errors in the caption.

- Figure 2: What is the horizontal line in panel 2e?

- Figure 7: it is difficult to identify the shading on the left as pink. Maybe choose a different color.

---

## Referee Comment (RC2) · Anonymous Referee #2 · 31 Aug 2019

The authors describe a spectroscopic technique (DOAS) for measuring emissions of NOx and SO2 using a mobile monitoring platform. Compared to satellite-based techniques, the method used here has advantages including higher spatial resolution and the possibility of making multiple measurements per day. The mobile DOAS technique is used to measure NOx and SO2 emissions from industrial sources in Sarnia, which is in southwestern Ontario close to the US border. An interesting feature of this work is the use of a NOx analyzer which provided measured NOx/NO2 ratios, facilitating the estimation of NOx emissions from NO2 column measurements.

The authors should address the following questions before the manuscript is published

in ACP.

Line 332: The Leighton ratio is calculated using measured NO and NO2 concentrations, but the NO2 measurement is likely to be biased high because of other nitrogen-containing pollutants such as peroxyacetyl nitrate, other organic nitrates, and nitrogen-containing acids that are included in the total NOx (and therefore also in the inferred NO2) concentration measurements. The authors conclude Leighton ratios provide evidence of peroxy radical-related deviations from the photo-stationary state relationship relating O3, NO, and and NO2 concentrations. Uncertainties in the NO2 measurement (calculated as NOx-NO) may also be a factor to consider.

Lines 423 and Line 570: fix "Canada and Canada" reference formatting errors

Line 656: please add a URL for this reference.

---

## Author Comment (AC1) · 25 Sep 2019

Response to Referee 1

Anonymous Referee #1

The manuscript by Davis et al. describes mobile multi-axis DOAS observations of industrial and urban emissions around the town of Sarnia, ON. The study is based on 3 days of observations of UV-vis absorption spectra in 30, 40, and 90 degree elevation viewing angles from a moving car downwind, and sometimes upwind, of the emission sources. The spectra were analyzed to retrieve NO2 and SO2 column densities. These were then converted into fluxes using 10m wind data. In addition, an in-situ NOx monitor installed on the vehicle was used to convert NO2 to NOx columns. The use of the NOx monitor is a nice touch as it reduces one of the main uncertainties when using DOAS for NOx flux measurements. The manuscript is thorough in its discussion of the methodology, and the authors should be commended for the detailed discussion of the uncertainties of their observations. The authors provide a preliminary comparison of their fluxes with those from a 2015 power plant emission inventory and a 2017 industrial facility emission inventory. The comparison is reasonable, considering the various uncertainties entering the determination of both emission rates. Overall, this is a very good manuscript, although I am wondering if it would have been better suited for Atmospheric Measurement Techniques, since most of the manuscript is dedicated to explaining the use of the MAX-DOAS technique to measure emissions. Maybe it would be worth discussing the emission results in more detail, i.e. a comparison with some other studies that show whether the agreement with the emission inventory is similar to other locations and/or addressing the overall question of the accuracy of emission inventories. This would make this study more valuable for ACP readers. Aside from these general comments I have a few other minor comments that should be addressed before the manuscript can be published in ACP.

Response: We thank Reviewer # 1 for their time.  At this time we would prefer to publish in ACP.  We think the subject topic meets the requirements for publication in ACP.  Most of the literature on mobile-MAX-DOAS appears in ACP, and the use of the NOx monitor is a minor incremental improvement to the method, not worthy of a methodology paper.  Our focus here is to alert the readers to the possibility of using mobile MAX-DOAS to measuring emissions from a small city, where the focus can be on measuring those emissions multiple times per day.  The results, as you say, are reasonable, however, a detailed discussion of the inventory comparison is premature as we would truly require more measurements to acquire better statistics.  Added to that, in a future study, additional wind measurements would be desirable above 10 m, to lower the uncertainty associated with the fluxes at elevation.

• I am a little confused about the $SO_2$ emission estimates. It seems those are only reported for day 1. What about the other days? Since NO2 data is available I assume that SO2 is available as well. If the SO2 was not above the detection limit it should be reported as upper limit emission estimate.

Response: Yes, you are correct, $SO_2$ was only reported for 1 day.  The reason is in fact that conditions were not optimum at other times for detection of $SO_2$, it was indeed below detection limits, making reporting not recommended.  If the uncertainty is so large that the VCDs and subsequent emissions are zero within error, then we take the approach that it should not be reported, similar to the principle of not reporting concentrations of species that are below detection limit.  This can occur for several reasons: i) $SO_2$ has an inherently lower differential cross section than $NO_2$, and it is being detected in a wavelength region where the actinic flux of scattered sunlight is much less than that for $NO_2$ ii) at times early or later in the day, actinic fluxes of UV radiation necessary for determination of $SO_2$ (e.g., 307-318nm) fall off much more quickly than visible light necessary for determination of $NO_2$ (410-435 nm) and ii) horizontal dilution conditions of $SO_2$ plumes can be variable.  Generally $SO_2$ is a weaker absorber than $NO_2$ so it frequently falls below detection limit when $NO_2$ does not. We have modified some text in the paper to clarify, including:

Section 2.2 - *The VCD of $SO_2$ was above detection limit on only two occasions in this study (both on Day 1), in contrast to $NO_2$.  The detection limit of $SO_2$ is higher than $NO_2$ for several reasons, first, it's differential cross*

*section is less than that of NO₂ and second, it's absorption features are in the UV wavelength region where scattered sunlight intensity is much less than that in the visible region. The fast measurements required in mobile DOAS also allow limited averaging of spectra compared to stationary measurements (Davis et al., 2019), where detection of industrial SO₂ plumes is easier. Therefore, SO₂ DSCDs were only above detection limits for Day 1 Routes 3 & 4 when the light levels were highest, and the vehicle observed the combined plumes of the largest SO₂ sources in the area.*

Section 3.4.1 *Apart from NOₓ, we were also able to estimate SO₂ emissions from the Sarnia urban/industrial region during one route when the SO2 DSCDs were detectable, Day 1 route 3 (Table 5)*

Section 3.6 *Very low background trace-gas levels also resulted in SO₂ DSCDs that were below detection limit most of the time, while being occasionally below detection limit for NO₂ (Fig. 2e). A spectrometer with higher sensitivity giving lower detection limits could solve this issue. Increased averaging of spectra would also improve detectability but at the expense of worse spatial resolution, unless measurements can be made at a slower driving speed.*

• Section 2.2: Was HCHO included in the fit of SO2? And more generally, why was HCHO not retrieved? HCHO column densities would provided information on VOC's that are discussed later in the manuscript.

Response: HCHO was not quantified, and was not included in the fit, as it is far below detectable in the current study, making virtually no impact on the residuals in the fits.  Differential cross sections of HCHO are even lower than those of SO₂, making it non-detectable at the high speed measurements required in mobile MAX-DOAS We expect the levels of HCHO to be generally low (a few ppb).

Additional text was added in section 2.2:

*Formaldehyde (HCHO) was not included in the fits for SO₂ as it was expected to be very low, and did not affect the residuals for the SO₂ fits.*

• Section 3.4.1: Are the emissions used for the NPRI comparison also scaled up to account for not measuring the entire plume?

Response: No.  We have added some clarifying text.  Only the flux integral needed to be scaled up by a factor of 2 since it only captured approximately half the plume.  Additional text in section 3.4.1:

*A preliminary estimation of the NOₓ and SO₂ emissions from the power-plants can be determined by scaling up the flux integral from the appropriate section of the East-West transect by a factor of two. While this is highly uncertain, we do this to make a first order estimate of the power plant plumes on the US side of the border.*

• Lines 270-271: ". . .thus offsetting."? I do not understand this sentence.

Response: we have reworded this…

*However, these errors are smaller than might be expected due to the presence of the error in both the numerator and the denominator of the ratio, NOₓ/NO₂ = (NO+NO₂)/NO₂, thus partially offsetting each other.*

• Lines 478 – 480, and other locations in the manuscript: Please clarify that one needs both the vertical wind profile, as well as the trace gas profile, for accurate flux determination. Wind profiles alone, while improving the calculation, are insufficient. Have you considered the change of wind direction in the boundary layer (Ekman spiral)?

Response: You are correct, for accurate flux, by the best of aircraft measurements, yes one needs the trace gas profile (concentration vs height) as well as the wind profile (wind vector vs height).  See Baray et al. (2018) for

example.  A trace gas profile by MAX-DOAS is possible in principle using optimal estimation and radiative transfer modelling but not without error, limited degrees of freedom and limited vertical resolution (see Davis et al., 2019). In this case we have the averaged or integrated trace gas profile, represented by the VCD (tropospheric).  To make an accurate flux estimate using this, presuming the VCD is accurate, one also needs an accurate wind vector that on average, represents the movement of the average column of air.  You are correct in pointing out that change in wind direction and speed in the column (wind shear or "Ekman spiral") introduces uncertainty and making estimates during such conditions should be avoided. We have added some clarifying text:

Section 1: *The major advantage of emissions estimates using aircraft measurements is that one can in principle fully characterize the vertical profile of trace gas concentration as well as the vertical profile of wind vectors for an accurate horizontal flux measurement downwind of a source (Baray et al., 2018; Gordon et al., 2015).*

and

*However one is still left with the uncertainty of the vertical profile of wind vector fields.*

and Section 3.6:

*Ideally accurate horizontal flux measurements would require knowledge of the vertical and horizontal profile of pollutant concentrations as well as vertical and horizontal profile of wind vectors.*

• Figures 2, 5, and 7: Add error bars, or at least list the errors in the caption.

Response: In order to keep the figure clear, we have added the estimated uncertainties to the figure captions in each case.

• Figure 2: What is the horizontal line in panel 2e?

Response: This is the zero line, yes it looked confusing. We have modified the figure, lining up the zero line on **both** the left and right axes.

• Figure 7: it is difficult to identify the shading on the left as pink. Maybe choose a different color.

Response: Yes, we agree, it was difficult to identify.  We have now darkened it in a new figure.

---

## Author Comment (AC2) · 25 Sep 2019

Response to Referee 2

**Anonymous Referee #2**

The authors describe a spectroscopic technique (DOAS) for measuring emissions of
NOx and SO2 using a mobile monitoring platform. Compared to satellite-based techniques,
the method used here has advantages including higher spatial resolution and
the possibility of making multiple measurements per day. The mobile DOAS technique
is used to measure NOx and SO2 emissions from industrial sources in Sarnia, which
is in southwestern Ontario close to the US border. An interesting feature of this work
is the use of a NOx analyzer which provided measured NOx/NO2 ratios, facilitating the
estimation of NOx emissions from NO2 column measurements.
The authors should address the following questions before the manuscript is published in ACP.

Response: We thank Reviewer # 2 for their time.

Line 332: The Leighton ratio is calculated using measured NO and $NO_2$ concentrations,
but the $NO_2$ measurement is likely to be biased high because of other nitrogen containing
pollutants such as peroxyacetyl nitrate, other organic nitrates, and nitrogen containing
acids that are included in the total NOx (and therefore also in the inferred
NO2) concentration measurements. The authors conclude Leighton ratios provide evidence
of peroxy radical-related deviations from the photo-stationary state relationship
relating O3, NO, and and NO2 concentrations. Uncertainties in the NO2 measurement
(calculated as NOx-NO) may also be a factor to consider.

Response – You are correct.  We did not address this for the Leighton ratio, although we did address the potential
bias in the NOx/NO2 ratio from these errors.  We have now addressed the potential bias in $\phi$ but it does not
change the interpretation.  Clarifying text:

Section 3.3: *Even if we consider a potential bias of + 20%  in the $NO_2$ measurements by the $NO_x$ analyzer for
reasons outlined in Section 3.2 (highly unlikely in a fresh $NO_x$ plume), a + 20% bias in the Leighton ratio would still
give ($\phi = 1.4$-$1.9$).*

**Footnote in Table 3:**  *Note that Leighton ratios, $\phi$, could be biased high by as much as +20% from the the $NO_z$ component of
$NO_y$ measured by the $NO_x$ analyzer, but likely much lower due to it being a fresh urban/industrial $NO_x$ plume.*

Lines 423 and Line 570: fix "Canada and Canada" reference formatting errors

Response - Fixed, should be ECCC.

Line 656: please add a URL for this reference.

Response - Fixed.

Additional references:

Davis, Z. Y. W., Frieβ, U., Strawbridge, K. B., Aggarwaal, M., Baray, S., Schnitzler, E. G., Lobo, A., Fioletov, V. E., Abboud, I.,
McLinden, C. A., Whiteway, J., Willis, M. D., Lee, A. K. Y., Brook, J., Olfert, J., O'Brien, J., Staebler, R., Osthoff, H. D., Mihele, C.,
and McLaren, R.: Validation of MAX-DOAS retrievals of aerosol extinction, $SO_2$ and $NO_2$ through comparison with lidar, sun
photometer, Active-DOAS and aircraft measurements in the Athabasca Oil Sands Region, Atmos. Meas. Tech. Discuss.,
https://doi.org/10.5194/amt-2019-296, in review, 2019.